# Induction Rather Than Imagination: Generative Zero-Shot Learning Via Inductive Variational Autoencoder

## Abstract

Remarkable progress in zero-shot learning (ZSL) has been achieved using generative models. However, existing generative ZSL methods merely generate (*imagine*) the visual features from scratch guided by the strong class semantic vectors annotated by experts, resulting in suboptimal generative performance and limited scene generalization. To address these and advance ZSL, we propose an inductive variational autoencoder for generative zero-shot learning, dubbed GenZSL. Mimicking human-level concept learning, GenZSL operates by *inducting* new class samples from similar seen classes using weak class semantic vectors derived from target class names (i.e., CLIP text embedding). To ensure the generation of informative samples for training an effective ZSL classifier, our GenZSL incorporates two key strategies. Firstly, it employs class diversity promotion to enhance the diversity of class semantic vectors. Secondly, it utilizes target class-guided information boosting criteria to optimize the model. Extensive experiments conducted on three popular benchmark datasets showcase the superiority and potential of our GenZSL with significant efficacy and efficiency over f-VAEGAN, e.g., 24.7% performance gains and more than $60\times$ faster training speed on AWA2. Codes are available at `https://anonymous.4open.science/r/GenZSL`.

## 1 Instruction

Zero-shot learning (ZSL) enables the recognition of unseen classes by transferring semantic knowledge from some seen classes to unseen ones [35; 27]. Recently, generative models such as generative adversarial networks (GANs) [17], variational autoencoders (VAEs) [25], and normalizing flows [16] have been successfully applied in ZSL, achieving significant performance improvements. These models synthesize images or visual features of unseen classes to alleviate the lack of samples for those classes [2; 52; 54; 7; 34; 8]. Given that GAN architectures can generate higher-quality visual sample features, there's a growing trend in synthesizing features using GANs [52; 54; 7; 34]. However, existing generative ZSL methods typically generate (*imagine*) visual features from scratch (e.g., Gaussian noises) guided by strong class semantic vectors [52; 54; 7; 34; 64; 11]. This approach often fails to produce reliable feature samples and generalize to various scene tasks, as illustrated in Figure 1 (a). The shortcomings arise from: i) the generator learning from scratch without sufficient data to capture the high-dimensional data distribution, and ii) the reliance on expert-annotated class semantic vectors, which are time-consuming and labor-intensive to collect for various scene generalizations. Hence, there's a pressing need to explore novel generative paradigms for the ZSL task.

Cognitive psychologist often frame the process of learning new concepts as "the problem of induction" [5; 1]. For instance, children typically induce novel concepts from a few familiar objects, guided by certain priors [45; 26]. Essentially, rich concepts can be induced "compositionally" from simpler primitives under a Bayesian criterion, and the model "learns to learn" by developing hierarchical priors that facilitate the learning of new concepts based on previous experiences with related concepts. These priors represent a learned inductive bias that abstracts the key regularities and dimensions of variation across both types of concepts and instances of a concept within a given domain. Following this paradigm, our objective is to devise a novel generative zero-shot learning (ZSL) model capable of generating (*inducing*) new/target classes based on samples from similar/referent seen classes. As

(a) Existing Generative ZSL       (b) Our GenZSL

Figure 1: Motivation illustration. (a) Existing generative ZSL methods merely generate (*imagine*) the visual features from scratch guided by the expert-annotated class semantic vectors, resulting in suboptimal generative performance and weak scene generalization. For example, the generator inevitably generates similar classes of "Zebra" or others, e.g., "Donkey". (b) Our GenZSL generates (*induces*) the reliable visual features of unseen classes from the similar seen classes with the clues of class semantic vector extracted by CLIP text encoder, e.g., from "Horse" to "Zebra".

illustrated in Figure 1 (b), our generative ZSL model can generate informative samples of new classes (e.g., "Zebra") by inducing them from referent seen classes (e.g., "Horse", "Tiger", and "Panda").

Indeed, there are two challenges in targeting this goal. Firstly, addressing the issue of weak class semantic vectors. These vectors, extracted from sources like the CLIP text encoder [37], often lack specific class information, such as attributes, compared to vectors annotated by experts. As a result, they may not effectively guide generative methods. Furthermore, these vectors can be misaligned in the vision-language space. For instance, the text embedding of a class name might be close to embeddings of unrelated classes but distant from image embeddings [21; 44]. How can we enhance the diversity of weak class semantic vectors to distinguish between various classes effectively, thereby avoiding the problem of generating visual features that are too similar to other classes? Secondly, ensuring that a novel generative method evolves samples of referent classes into target classes with the guidance of weak class semantic vectors is equally challenging. This involves transforming samples of seen classes into samples that accurately represent unseen classes, guided only by the limited information provided by weak class semantic vectors. How can we achieve this transformation reliably and effectively within a generative ZSL framework?

To guide the induction towards creating informative samples for training effective ZSL classifiers, we propose a novel inductive variational autoencoder for generative ZSL, namely **GenZSL**. Specifically, GenZSL considers two criteria, i.e., class diversity promotion and target class-guided information boosting. In addressing the first criterion, we reduce redundant information from class semantic vectors by eliminating their major components. This process enables all class semantic vectors to become nearly perpendicular to each other but keep the origin relationships between all classes, thus enhancing the diversity among them. For the second one, we design a target class-guided information boosting loss to guide GenZSL to synthesize the visual features belonging to target classes.

Our main contributions are summarized in the following:

**i)** We propose an induction-based GenZSL for generative ZSL, which can synthesize the samples of unseen classes based on the weak class semantic vectors inducting from the similar seen classes. To the best of our knowledge, GenZSL stands as the first inductive generative method, offering a unique and innovative solution distinct from existing approaches.

**ii)** We enable GenZSL to synthesize informative samples by improving class diversity between various class semantic vectors and designing the target class-guided information boosting criteria.

**iii)** We conduct extensive experiments on three wide-use ZSL benchmarks (e.g., CUB [49], SUN [36], and AWA2 [53]), results demonstrate the significant efficacy and efficiency over the existing ZSL methods, e.g., 24.7% performance gains and more than $60\times$ faster training speed on AWA2. More importantly, our GenZSL can be flexibly extended on various scene tasks without the guidance of expert-annotated attributes.

## 2    RELATED WORK

**Zero-Shot Learning.**    Zero-shot learning is proposed to tackle the classification problem when some classes are unknown. To recognize the unseen classes, the side-information/semantic (e.g., attribute descriptions [28], DNA information [4]) is utilized to bridge the gap between seen and unseen classes. As such, the key task of ZSL is to conduct effective interactions between visual and semantic domains. Typically, there are two methodologies to target on this goal, i.e., embedding-based methods that learn visual→semantic mapping [51; 56; 63; 47; 19], and generative methods that learn semantic→visual mapping [54; 7; 23; 64; 13]. Considering the semantic representations, embedding-based methods focus recently on learning the region-based visual features rather than the holistic visual features [22; 56; 9; 10; 12]. Since these methods learn the ZSL classifier only on seen classes, inevitably resulting in the models overfitting to seen classes. To tackle this challenge, generative ZSL methods employ the generative models (e.g., VAE, and GAN) to generate the unseen features for data augmentation, and thus ZSL is converted to a supervised classification task. As such, the generative ZSL methods have shown significant performance and become very popular recently. Furthermore, Li *et al.* [29] introduces Stable Diffusion to perform zero-shot classification without any additional training by leveraging the ELBO as an approximate class-conditional log-likelihood.

However, existing generative ZSL methods simply imagine the visual feature from a Gaussian distribution with the guidance of a strong class semantic vector. Thus, they are limited in i) there lacks enough data for training a generative model to learn the high-dimension data distribution, resulting in undesirable generation performance; ii) they rely on the strong condition guidance (e.g., expert-annotated attributes) for synthesizing target classes, so they cannot easily generalize to various scenes. As such, we propose a novel generative method to create informative samples of unseen classes for advancing ZSL via induction rather than imagination.

**Generative Model for Data Augmentation.**    Synthesizing new data using a generative model for data augmentation is a promising direction [61; 24; 20]. Many recent studies [3; 18; 57; 50] explored generative models to generate new data for model training. However, these methods fail to ensure that the synthesized data bring sufficient new information and accurate labels for the target small datasets. Because they imagine the new data from scratch (e.g., Gaussian distribution), which is infeasible with very limited/diverse training data. Zhang *et al.* [60] introduce GIF to expanding small-scale datasets with guided imagination using pre-trained large-scale generative models, e.g., Stable Diffusion [39] or DALL-E2 [38]. Although GIF can expand a small dataset into a larger labeled one in a fully automatic manner without involving human annotators, it requires anchor samples for imagination. As such, these imagination-based generative models are not feasible for ZSL tasks. In contrast, we introduce a novel generative method to synthesize new informative data for ZSL via induction inspired by the human perception process [5; 1].

## 3    INDUCTIVE VARIATIONAL AUTOENCODER FOR ZSL

**Problem Setting.**    The problem setting of ZSL and notations are defined in the following. Assume that data of seen classes $\mathcal{D}^s = \{(x_i^s, y_i^s)\}$ has $C^s$ classes, where $x_i^s \in \mathcal{X}$ denotes the $i$-th visual feature extracted from the CLIP visual encoder [37], and $y_i^s \in \mathcal{Y}^s$ is the corresponding class label. $\mathcal{D}^s$ is further divided into training set $\mathcal{D}_{tr}^s$ and test set $\mathcal{D}_{te}^s$ following [53]. The unseen classes $C^u$ has unlabeled samples $\mathcal{D}_{te}^u = \{(x_i^u, y_i^u)\}$, where $x_i^u \in \mathcal{X}$ are the visual samples of unseen classes, and $y_i^u \in \mathcal{Y}^u$ are the corresponding labels. A set of class semantic vectors of the class $c \in \mathcal{C}^s \cup \mathcal{C}^u = \mathcal{C}$ are extracted from CLIP text encoder, defined as $z^c$. In the conventional zero-shot learning (CZSL) setting, we learn a classifier only classifying unseen classes, i.e., $f_{CZSL} : X \to Y^U$, while we learn a classifier for both seen and unseen classes in the generalized zero-shot learning (GZSL) setting, i.e., $f_{GZSL} : X \to Y^U \cup Y^S$.

**Pipeline Overview.**    To enable the generative ZSL method to synthesize high-quality visual features with good scene generalization, we propose an inductive variational autoencoder for ZSL (namely GenZSL). Towarding to creating informative new samples for unseen classes, GenZSL considers two important criteria, i.e., class diversity promotion and target class-guided information boosting. As shown in Fig. 2, GenZSL first takes class diversity promotion to reduce the redundant information from class semantic vectors by removing their major components, enabling

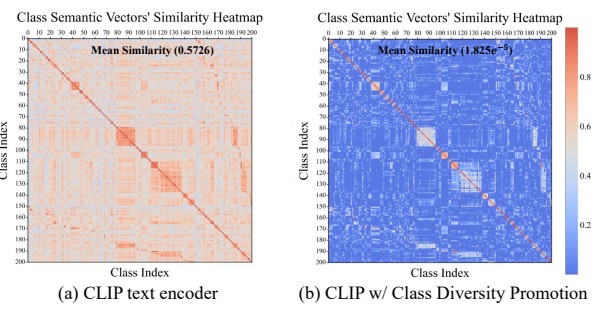

Figure 2: Pipeline of our GenZSL. GenZSL first takes class diversity promotion to reduce the redundant information from class semantic vectors, and to improve the identity for all class semantic vectors. Then, it employs a semantically similar sample selection module to select the top-$k$ referent class from the seen classes for each target class as training inputs. Based on the referent samples, GenZSL learns an inductive variational autoencoder to create the new informative feature samples for unseen classes via induction optimized by target class-guided information boosting criteria.

all class semantic vectors nearly perpendicular to each other. Based on the refined class semantic vectors, GenZSL employs a semantically similar sample selection module to select the top-$k$ referent class from the seen classes for each target class. Subsequently, GenZSL learns the inductive variational autoencoder (IVAE) with the Kullback-Leibler divergence (KL) loss, target class reconstruction loss, and target class-guided information boosting loss, which ensures GenZSL inducts the target class samples from their similar class samples. After training, GenZSL takes IVAE to synthesize visual features of unseen classes to learn a supervised classifier.

### 3.1 CLASS DIVERSITY PROMOTION

To avoid the ZSL model relying on the expert-annotated class semantic vectors, we adopt CLIP [37] text encoder to extract the class semantic vectors, i.e., text embedding of the class names. However, we observed that the CLIP text encoder fails to capture discriminative class information, especially on fine-grained datasets. As shown in Fig. 3(a), the class semantic vectors have high similarity with other classes, that is, all class semantic vectors are highly adjacent to ones of other classes. If we directly take such class semantic vectors as conditions to guide GenZSL, it inevitably causes the synthesized visual features confusion as the class semantic vectors with limited diversity.

Figure 3: Class semantic vectors' similarity heatmaps are extracted by CLIP text encoder and CLIP with class diversity promotion on the CUB dataset. The similarity heatmaps on SUN and AWA2 are presented in Appendix B.

As such, we introduce class diversity promotion (CDP) to improve the diversity of class semantic vectors. CDP reduces the redundant information from class semantic vectors by removing their major components, enabling all class semantic vectors nearly perpendicular to each other but to keep the original class relationships. Specifically, we take Singular Value Decomposition to get the orthonormal basis of the span of class semantic vectors $Z = [z^1, z^2, \cdots, z^C]$, i.e., $U, S, V = svd(Z)$, where $U = [e^1, e^2, \cdots, e^C]$ is the orthonormal basis. As suggested in Principal Component Analysis, the first dimension $e^1$ of the outer-space basis $U$ will be the major component, which overlaps on most class semantic vectors $[z^1, z^2, \cdots, z^C]$. We directly remove the major component $e^1$ to define the new projection matrix $P = U^{'}U^{'\top}$ with $U^{'} = [e^2, e^3, \cdots, e^C]$. Accordingly, we obtain the refined class semantic vectors, formulated as:

$$\tilde{Z} = P \cdot Z = \{\tilde{z}^1, \tilde{z}^2, \cdots, \tilde{z}^C\} \tag{1}$$

As shown in Fig. 3(b), we make the refined class semantic vectors nearly perpendicular to each other, such as the mean similarity between various classes drops from $0.5726$ to $1.825e^{-5}$ on the CUB dataset. As such, the refined class semantic vectors will be the significant conditions for induction.

## 3.2 SEMANTICALLY SIMILAR SAMPLE SELECTION

In this paper, we are interested in semantically similar samples as they can serve as reliable known data for inducing new samples of other similar classes. Specifically, we select the semantically similar samples in seen classes (defined a referent class samples) with respect to the target seen/unseen classes $c^{target}$ during training/testing, respectively. According to the cosine similarity, we define similar samples as the referent ones whose class semantic vectors $\tilde{z}^{c^s}$ is top-$k$ closed to the target class semantic vectors $\tilde{z}^{target}$, formulated as:

$$c^{refer} = \arg \max_{top-k(c^s)} \frac{\tilde{z}^{target} \times \tilde{z}^{c^s}}{\|\tilde{z}^{target}\| \cdot \|\tilde{z}^{c^s}\|}, \tag{2}$$

where $k$ is the number of referent classes with respect to the corresponding target classes. Accordingly, we can obtain a set of referent samples to the target seen/unseen classes from seen classes for training/testing, respectively.

## 3.3 INDUCTIVE VARIATIONAL AUTOENCODER

**Network Components.**  Our GenZSL aims to generate informative new samples for novel classes by inducing from seen classes. To achieve this, we devise a novel generative model called the inductive variational autoencoder (IVAE). We formulate the induction of new samples for target classes $\hat{x}$ from reference samples $x^{refer}$ as $\hat{x} = IVAE(x^{refer} + o, \tilde{z}^{target})$, where $o$ represents the perturbation applied to $x^{refer}$ to enable IVAE to variationally generate $\hat{x}$ distinct from $x^{refer}$.

Specifically, IVAE consists of an inductive encoder (IE) and an inductive decoder (ID). The IE and ID are the Multi-Layer Perceptron (MLP) networks. The IE encodes the referent samples $x^{refer}$ into latent space $o$ conditioned by the target class semantic vectors $\tilde{z}^{target}$, i.e., $o = \delta \cdot \mathcal{N}(0, 1) + \mu$, where $\mu, \delta = IE(x^{refer}, \tilde{z}^{target})$. Subsequently, The ID further comprises hidden layers with a progressively larger number of nodes that decode the latent features to be a reconstruction of the target classes samples $x^{target}$ guided by $\tilde{z}^{target}$, formulated as $\hat{x} = ID(o, \tilde{z}^{target})$. This is different to VAE which ultimately reconstructs the data back to its original input $x^{refer}$.

**Network Optimization.**  Similar to the conditional VAE [43], our IVAE includes the KL loss $\mathcal{L}_{KL}$ and the target class reconstruction loss $\mathcal{L}_{TR}$, formulated as:

$$\begin{aligned} \mathcal{L}_{IVAE} &= \mathcal{L}_{KL} - \mathcal{L}_{TR} \\ &= KL(q(o \mid x, \tilde{z}^{target}) \| p(o \mid \tilde{z}^{target})) - \mathbb{E}_{q(o|x^{refer}, \tilde{z}^{target})}[\log p(x^{target} \mid o, \tilde{z}^{target})], \end{aligned} \tag{3}$$

where $q(o \mid x, \tilde{z}^{target})$ is modeled by $IE(x^{refer}, \tilde{z}^{target})$, $p(o \mid \tilde{z}^{target})$ is assumed to be $\mathcal{N}(0, 1)$, and $p(x^{target} \mid o, \tilde{z}^{target})$ is represented by $ID(o, \tilde{z}^{target})$. Essentially, $\mathcal{L}_{TR}$ towards the target class-guided information boosting criteria in vision-level, encouraging IVAE to synthesize high-quality target class samples.

To ensure IVAE evolves the referent samples to belong to target classes, GenZSL further employs a target class-guided information boosting loss $\mathcal{L}_{Boost}$ for optimization. Considering CLIP's full prior knowledge, $\mathcal{L}_{Boost}$ aims to improve the information entropy between the synthesized visual features of target classes $\hat{x}^{target}$ and their corresponding class semantic vectors $\tilde{z}^{target}$, formulated as:

$$\mathcal{L}_{Boost} = -\frac{\exp{(< \hat{x}^{target}, \tilde{z}^{target} > /\tau)}}{\sum_{j=1}^{C^s} \exp{(< \hat{x}^{target}, \tilde{z}^{target} > /\tau)}}, \tag{4}$$

where $\tau$ is the temperature parameter and set to 0.07. Indeed, $\mathcal{L}_{Boost}$ and $\mathcal{L}_{TR}$ cooperatively ensure IVAE to synthesize desirable target class samples from semantic- and vision-level, respectively.

As such, the total optimization loss function can be written as:

$$\mathcal{L}_{total} = \mathcal{L}_{IVAE} + \lambda \mathcal{L}_{Boost}, \tag{5}$$

where $\lambda$ is a weight to control the $\mathcal{L}_{Boost}$, enabling model optimization to be more effective.

## 3.4 ZSL CLASSIFICATION

After training, we first take the pre-trained IVAE to synthesize visual features for unseen classes:

$$\hat{x}^u = ID(o, \tilde{z}^{c^u}), \quad \text{where} \quad o = \delta \cdot \mathcal{N}(0,1) + \mu, and \quad \mu, \delta = IE(x^{refer}, \tilde{z}^{c^u}). \tag{6}$$

Different from the standard VAEs that synthesize samples from scratch (e.g., Gaussian noise), we synthesize the visual features of unseen classes inducting from referent seen class samples and take Gaussian noise as variations. As such, our GenZSL can more easily create informative new samples for unseen classes.

Then, we take the synthesized unseen visual features (and the real visual features of seen classes $x^s \in \mathcal{D}^s_{tr}$) to learn a classifier (e.g., softmax), i.e., $f_{czsl} : \mathcal{X} \to \mathcal{Y}^s \cup \mathcal{Y}^u$ in the CZSL setting (and $f_{gzsl} : \mathcal{X} \to \mathcal{Y}^s \cup \mathcal{Y}^u$ in the GZSL setting). Once the classifier is trained, we use the real sample in the test set $\mathcal{D}^u_{te}$ to test the model further. The details of the testing process are shown in Appendix A.

## 4 EXPERIMENTS

**Datasets.** We evaluate our GenZSL on three well-known ZSL benchmark datasets, i.e., two fine-grained datasets ( CUB [49] and SUN [36]) and one coarse-grained dataset (AWA2 [53]). CUB has 11,788 images of 200 bird classes (seen/unseen classes = 150/50). SUN contains 14,340 images of 717 scene classes (seen/unseen classes = 645/72). AWA2 consists of 37,322 images of 50 animal classes (seen/unseen classes = 40/10).

**Evaluation Protocols.** During testing, we adopt the unified evaluation protocols following [53]. The top-1 accuracy of the unseen class (denoted as $acc$) is used for evaluating the CZSL performance. In the GZSL setting, the top-1 accuracy on seen and unseen classes is adopted, denoted as $S$ and $U$, respectively. Meanwhile, their harmonic mean (defined as $H = (2 \times S \times U)/(S + U)$) is a better protocols in the GZSL.

**Implementation Details.** We use the training splits proposed in [52]. Meanwhile, the visual features with 512 dimensions are extracted from the CLIP vision encoder [37]. The IE and ID are the MLP networks. The specific network settings are $fc(512) - fc(1024) - fc(2048) - ReLu$ and $fc(512) - fc(1024) - fc(2048) - ReLu - fc(512)$ for IE and ID, respectively. We synthesize 1600, 800, and 5000 features per unseen class to train the classifier for CUB, SUN, and AWA2 datasets, respectively. We empirically set the loss weight $\lambda$ as 0.1 for CUB and AWA2, and 0.001 for SUN. The top-2 similar classes serve as the referent classes for inductions on all datasets. Furthermore, to enlarge the reference of the referent samples for effective model training, we take mixup technique [59] to randomly fuse the samples of various referent classes for data augmentation, i.e., $x^{refer} = 0.8 \cdot x^{c^{top-1}} + 0.2 \cdot x^{c^{top-2}}$. All experiments are performed on a single NVIDIA TITAN X with 11G memory. We employ Pytorch to implement our experiments.

### 4.1 COMPARISONS WITH STATE-OF-THE-ART METHODS

We first compare our GenZSL with the various imagination-based generative ZSL methods (e.g., VAE [40; 8], GAN [52; 30; 46], VAEGAN [54; 11], normalizing flow [41], and gaussian feature generator [6]) under the CZSL. Table 1 shows the evaluation results on three datasets. Our GenZSL consistently achieves the best results with the $acc$ values of 63.3%, 73.5%, and 92.2% on CUB, SUN, and AWA2, respectively. Notably, our GenZSL obtains the performance gains by 20.3% at least on AWA2 over the imagination-based generative ZSL methods. These competitive results demonstrate the superiority and potential of our induction-based

Table 1: Comparison with generative ZSL methods on three datasets under CZSL setting.

| Methods | CUB | SUN | AWA2 |
|---|---|---|---|
| | acc | acc | acc |
| CLSWGAN [52] | 57.3 | 60.8 | 68.2 |
| f-VAEGAN [54] | 61.0 | 64.7 | 71.1 |
| CADA-VAE [40] | 59.8 | 61.7 | 63.0 |
| LisGAN [30] | 58.8 | 61.7 | 70.6 |
| IZF-NBC [41] | 59.6 | 63.0 | 71.9 |
| LsrGAN [46] | 60.3 | 62.5 | 66.4 |
| HSVA [8] | 62.8 | 63.8 | 70.6 |
| GG [6] | 60.3 | 62.7 | 70.1 |
| f-VAEGAN+DSP [11] | 62.8 | 68.6 | 71.6 |
| **GenZSL (Ours)** | **63.3** | **73.5** | **92.2** |

Table 2: State-of-the-art comparisons for ZSL methods on CUB, SUN, and AWA2 under GZSL settings. Embedding-based methods are categorized as †, and generative methods are categorized as ‡. ∗ denotes ZSL methods using attribute features to refine visual features. The best and second-best results are marked in **Red** and **Blue**, respectively.

| | Methods | Venue | CUB | | | SUN | | | AWA2 | | |
|---|---|---|---|---|---|---|---|---|---|---|---|
| | | | U | S | H | U | S | H | U | S | H |
| † | SGMA [63] | NeurIPS'19 | 36.7 | 71.3 | 48.5 | – | – | – | 37.6 | 87.1 | 52.5 |
| | AREN [55] | CVPR'19 | 38.9 | **78.7** | 52.1 | 19.0 | 38.8 | 25.5 | 15.6 | **92.9** | 26.7 |
| | CRnet [58] | ICML'19 | 45.5 | 56.8 | 50.5 | 34.1 | 36.5 | 35.3 | 52.6 | 78.8 | 63.1 |
| | APN∗ [56] | NeurIPS'20 | 65.3 | 69.3 | 67.2 | 41.9 | 34.0 | 37.6 | 56.5 | 78.0 | 65.5 |
| | DAZLE∗ [22] | CVPR'20 | 56.7 | 59.6 | 58.1 | **52.3** | 24.3 | 33.2 | 60.3 | 75.7 | 67.1 |
| | CN [42] | ICLR'21 | 49.9 | 50.7 | 50.3 | 44.7 | 41.6 | 43.1 | 60.2 | 77.1 | 67.6 |
| | TransZero∗ [9] | AAAI'22 | **69.3** | 68.3 | **68.8** | **52.6** | 33.4 | 40.8 | 61.3 | 82.3 | 70.2 |
| | MSDN∗ [10] | CVPR'22 | **68.7** | 67.5 | **68.1** | 52.2 | 34.2 | 41.3 | 62.0 | 74.5 | 67.7 |
| | I2DFormer [32] | NeurIPS'22 | 35.3 | 57.6 | 43.8 | – | – | – | **66.8** | 76.8 | 71.5 |
| | I2MVFormer-Wiki [33] | CVPR'23 | 32.4 | 63.1 | 42.8 | – | – | – | 66.6 | 82.9 | 73.8 |
| | ICIS [15] | ICCV'23 | 45.8 | **73.7** | 56.5 | 45.2 | 25.6 | 32.7 | 35.6 | **93.3** | 51.6 |
| ‡ | CLSWGAN [52] | CVPR'18 | 43.7 | 57.7 | 49.7 | 36.6 | 42.6 | 39.4 | 52.1 | 68.9 | 59.4 |
| | f-VAEGAN [54] | CVPR'19 | 48.7 | 58.0 | 52.9 | 45.1 | 38.0 | 41.3 | 57.6 | 70.6 | 63.5 |
| | LisGAN [30] | CVPR'19 | 46.5 | 57.9 | 51.6 | 42.9 | 37.8 | 40.2 | 52.6 | 76.3 | 62.3 |
| | LsrGAN [46] | ECCV'20 | 48.1 | 59.1 | 53.0 | 44.8 | 37.7 | 40.9 | 54.6 | 74.6 | 63.0 |
| | AGZSL [14] | ICLR'21 | 48.3 | 58.9 | 53.1 | 29.9 | 40.2 | 34.3 | 65.1 | 78.9 | 71.3 |
| | HSVA [8] | NeurIPS'21 | 52.7 | 58.3 | 55.3 | 48.6 | 39.0 | 43.3 | 59.3 | 76.6 | 66.8 |
| | FREE+ESZSL [64] | ICLR'22 | 51.6 | 60.4 | 55.7 | 48.2 | 36.5 | 41.5 | 51.3 | 78.0 | 61.8 |
| | CLSWGAN + DSP [11] | ICML'23 | 51.4 | 63.8 | 56.9 | 48.3 | **43.0** | **45.5** | 60.0 | 86.0 | 70.7 |
| | **GenZSL** | Ours | 53.5 | 61.9 | 57.4 | 50.6 | **43.8** | **47.0** | **86.1** | 88.7 | **87.4** |

Table 3: Results of ablation study for our GenZSL on CUB and AWA2.

| Methods | CUB | | | | AWA2 | | | |
|---|---|---|---|---|---|---|---|---|
| | CZSL | GZSL | | | CZSL | GZSL | | |
| | acc | U | S | H | acc | U | S | H |
| GenZSL w/o CDP | 60.9 | 48.2 | 64.6 | 55.2 | 90.7 | 82.3 | 87.9 | 85.0 |
| GenZSL w/o $\mathcal{L}_{TR}$ | 48.3 | 20.1 | 37.5 | 26.2 | 87.5 | 39.9 | 83.1 | 53.9 |
| GenZSL w/o $\mathcal{L}_{Boost}$ | 61.1 | 47.7 | 66.4 | 55.5 | 90.5 | 75.3 | 91.4 | 82.6 |
| GenZSL w/o CDP&$\mathcal{L}_{Boost}$ | 60.0 | 42.5 | 69.3 | 52.7 | 87.7 | 89.0 | 75.3 | 81.6 |
| GenZSL (full) | 63.3 | 53.5 | 61.9 | 57.4 | 92.2 | 86.1 | 88.7 | 87.4 |

generative method, which significantly synthe-
sizes informative new samples for unseen classes.

Besides evaluating the CZSL performance, we also take our GenZSL to compare with the state-of-the-art ZSL methods under the GZSL setting, including the embedding-based methods and generative methods. Results are shown in Table 2. Compared to the embedding-based methods, our GenZSL achieves the best performance on harmonic mean on SUN and AWA2, and competitive results on CUB. It's worth noting that ZSL methods using attribute features to refine visual features can significantly improve their performances on CUB, e.g., APN [56], TransZero [9]. Because they can localize the specific attributes for visual representations. When taking our GenZSL to compare with the imagination-based generative methods, GenZSL performs best results of $H$=57.4%, $H$=47.0% and $H$=87.4% on CUB, SUN and AWA2, respectively. Notably, our GenZSL relies solely on weak class semantic vectors, while the compared methods utilize strong ones annotated by experts. This indicates that GenZSL is more adaptable to generalizing across various scenes. These results consistently demonstrate our induction-based GenZSL is a desirable generative paradigm for ZSL.

## 4.2 ABLATION STUDY

**Various Model Components of Our GenZSL.** To gain further insights into GenZSL, we conducted ablation studies to evaluate the effect of various model components, specifically class diversity promotion (CDP), the target class reconstruction loss $\mathcal{L}_{TR}$, and the target class-guided information boosting loss $\mathcal{L}_{Boosting}$, on the CUB and AWA2 datasets. The ablation results are summarized in Table 3. When GenZSL lacks CDP to consider class diversity criteria, there is a notable degradation in performance. This is attributed to the inability of class semantic vectors extracted from the CLIP text encoder to capture discriminative class information, resulting in weak diversity among class semantic vectors. Moreover, if GenZSL does not incorporate $\mathcal{L}_{TR}$ for target class information boosting, there is a significant drop in performance, with the harmonic mean decreasing by 30.8% and 33.5% on CUB

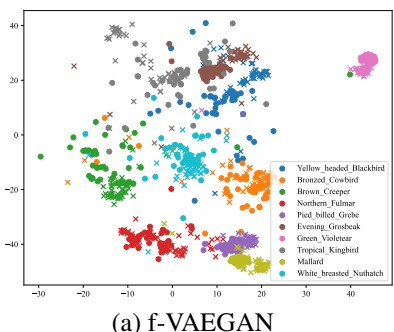
(a) f-VAEGAN

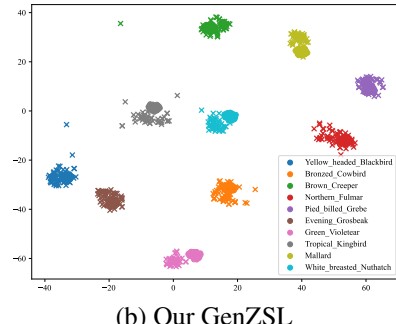
(b) Our GenZSL

Figure 4: Qualitative evaluation with t-SNE visualization. The sample features from f-VAEGAN [54] are shown on the left, and from our GenZSL are shown on the right. We use 10 colors to denote randomly selected 10 classes from CUB. The "×" and "○" are denoted as the real and synthesized sample features, respectively. The synthesized sample features and the real features distribute differently on the left while distributing similarly on the right. The t-SNE visualization on the SUN and AWA2 datasets is shown in Appendix D.

and AWA2, respectively. These findings underscore the importance of $\mathcal{L}_{TR}$ as a fundamental loss for target class-guided information boosting, ensuring that our IVAE accurately induces referent samples to target class samples. Furthermore, $\mathcal{L}_{Boosting}$ enhances the induction process at the semantic level, complementing $\mathcal{L}_{TR}$. Overall, these results demonstrate the effects of various components of GenZSL and underscore the significance of the two criteria for induction.

**Various Models with Weak Class Semantic Vectors.** We conducted a comparative analysis of various models utilizing weak class semantic vectors extracted from the CLIP text encoder. These models include large-scale visual-language-based ZSL methods such as CLIP [37], CoOp [62], and CoOp + SHIP [48], as well as classical generative ZSL methods like f-VAEGAN [54] and TF-VAEGAN [34]. The results are presented in Table 4. Compared to large-scale visual-language methods, our GenZSL demonstrates substantial improvements, in-

Table 4: Results of various models using weak class semantic vectors as side-information on CUB.

| Methods | CUB | | |
|---|---|---|---|
| | U | S | H |
| CLIP [37] | 55.2 | 54.8 | 55.0 |
| CoOp [62] | 49.2 | 63.8 | 55.6 |
| CoOp + SHIP [48] | 55.3 | 58.9 | 57.1 |
| f-VAEGAN [54] | 22.5 | 82.2 | 35.3 |
| TF-VAEGAN [34] | 21.1 | 84.4 | 34.0 |
| **GenZSL (Ours)** | 53.5 | 61.9 | 57.4 |

dicating the effectiveness of our inductive generative paradigm as a desirable ZSL model. When imagination-based generative ZSL methods utilize weak class semantic vectors as side information, GenZSL achieves significant performance gains, with a minimum increase of 22.1% in harmonic mean over these methods. Additionally, we observed that when imagination-based generative ZSL methods use weak class semantic vectors, their performances experience more significant drops compared to when they utilize strong class semantic vectors. For instance, the harmonic mean of f-VAEGAN decreases from 52.9% to 35.3%. These findings highlight the superiority of our induction-based generative method over imagination-based approaches in ZSL, as it can synthesize high-quality sample features for unseen classes with feasible scene generalization. Moreover, our work bridges the gap between large-scale visual-language ZSL methods and classical ZSL methods, leveraging the advantages of both approaches to achieve improved performance in ZSL tasks. More discussions are in Appendix C.

### 4.3 QUALITATIVE EVALUATION

We conducted a qualitative evaluation to intuitively showcase the performance of imagination-based generative ZSL methods (e.g., f-VAEGAN [54]) and our induction-based approach (GenZSL). The t-SNE visualization [31] of real and synthesized sample features is presented in Fig. 4. We randomly selected 10 classes from CUB and visualized the sample features generated by f-VAEGAN and GenZSL. Fig. 4(a) illustrates that sample features synthesized by f-VAEGAN and real features exhibit significant differences, indicating that the synthesized visual features may not facilitate

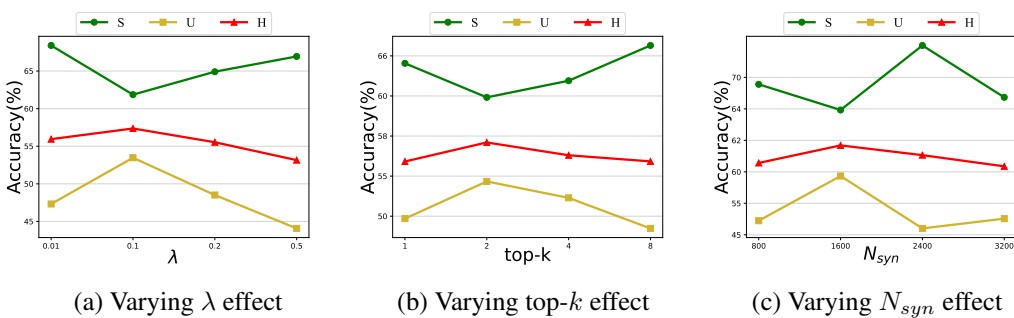

(a) Varying $\lambda$ effect      (b) Varying top-$k$ effect      (c) Varying $N_{syn}$ effect

Figure 6: Hyper-parameter analysis. We show the performance variations on CUB by adjusting the value of loss weight $\lambda$ in (a), the number of the top referent classes top-$k$ in (b), and the number of synthesized samples of each unseen class $N_{syn}$ in (c).

reliable classification for ZSL. In contrast, Fig. 4(b) demonstrates that our GenZSL synthesizes informative samples for unseen classes that closely match real sample features. This visualization confirms that GenZSL is a desirable generative ZSL model, and the induction-based generative paradigm holds value for ZSL tasks.

### 4.4 INDUCTION *vs* IMAGINATION

We analyze the efficiency and efficacy of induction-based generative ZSL (e.g., our GenZSL) and imagination-based generative ZSL (e.g., f-VAEGAN [54]) on AWA2. Results are shown in Fig. 5. We find that our GenZSL eases the optimization by providing faster convergence at the early stage, while f-VAEGAN towards convergence slowly. For example, GenZSL achieves the best GZSL performance with a remarkable $\geq 60\times$ acceleration in training speed than f-VAEGAN. Meanwhile, our GenZSL obtains better performance both in the GZSL and CZSL settings than f-VAEGAN. These demonstrate the efficiency and efficacy of our GenZSL and the great potential of the induction-based generative paradigm.

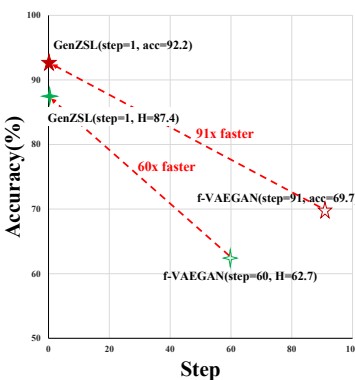

Figure 5: Induction *vs* Imagination on AWA2 dataset.

### 4.5 HYPER-PARAMETER ANALYSIS.

We analyze the effects of different hyper-parameters of our GenZSL on the CUB dataset. These hyper-parameters include the loss weight $\lambda$ in Eq. 5, the number of the top referent classes top-$k$, and the number of synthesized samples for each unseen class $N_{syn}$. Fig. 6 shows the CZSL and GZSL performances using different hyper-parameters. In (a), the results indicate that GenZSL is robust to varying values of $\lambda$ and achieves good performance when $\lambda$ is relatively small (i.e., $\lambda = 0.1$). This is because $\mathcal{L}_{Boost}$ is a semantic-level toward target class-guided information boosting criteria, which is a supplement to the vision-level one (e.g., $\mathcal{L}_{LR}$). In (b), we evaluate the top similar classes as referent classes varying $k = \{1, 2, 4, 8\}$. We find that our GenZSL uses the $top - 2$ referent classes to obtain better performance, which brings the mixup technique for data augmentation. In (c), our GenZSL is shown robust to $N_{syn}$ when it is not set in a large number. The $N_{syn}$ can be set as 1600 to balance between the data amount and the ZSL performance. Overall, Fig. 6 shows that our GenZSL is robust to overcome hyper-parameter variations. The hyper-parameter analysis on SUN and AWA2 are presented in Appendix E. Accordingly, we empirically set these hyper-parameters $\{\lambda, k, N_{syn}\}$ as $\{0.1, 2, 1600\}$, $\{0.001, 2, 800\}$ and $\{0.1, 2, 5000\}$ for CUB, SUN and AWA2, respectively.

## 5 LIMITATION DISCUSSION

The potential limitations of our GenZSL includes:

- If there lacks enough similar seen classes as reference, IVAE may need more learning time to evolve the referent samples to be target samples;
- The CLIP text embedding of class name lacks informative class information, which hampers the knowledge transfer of GenZSL.

# 6 CONCLUSION

In this work, we propose an inductive variational autoencoder as a new generative model for zero-shot learning, namely GenZSL. Inspired by human perception, GenZSL operates on an induction-based approach to synthesize informative and high-quality sample features for unseen classes. To achieve this, we introduce class diversity promotion to enhance the diversity and discrimination of class semantic vectors. Additionally, we design two losses targeting the criteria of target class-guided information boosting to optimize the model. Through qualitative and quantitative analyses, we demonstrate that GenZSL consistently outperforms existing generative ZSL methods in terms of efficacy and efficiency. We hope that our induction-based generative method offers new insights into zero-shot learning and other generation tasks, paving the way for further advancements in these areas.

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

APPENDIX

Appendix organization:

## A  TESTING PROCESS OF GENZSL

We present the testing process of GenZSL in Fig. 7. Different to the standard VAE that samples the new data from Gaussian noise, our GenZSL inducts the informative new sample features for unseen classes from the similar seen classes and takes Gaussian noises to enable IVAE to synthesize variable and diverse samples. Then, we take the synthesized unseen class samples $\hat{x}^u$ to learn a supervised classifier (e.g., softmax), which is used for ZSL evaluation further.

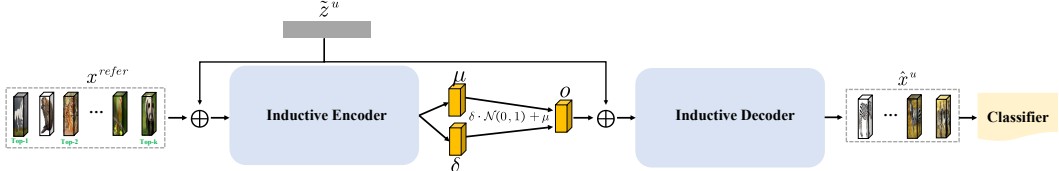

Figure 7: Testing process of GenZSL.

## B  CLASS SEMANTIC VECTORS' SIMILARITY HEATMAPS

We show the lass semantic vectors' similarity heatmaps of SUN and AWA2 in Fig. 8. Results show that our CDP effectively improves the discrimination and diversity for class semantic vectors, avoiding the confusion of synthesized visual features between various classes. For example, the mean similarity of class semantic vectors on AWA2 is reduced from 0.7609 to 0.0005. As such, the class semantic vectors served as a distinct conditions for effective generation.

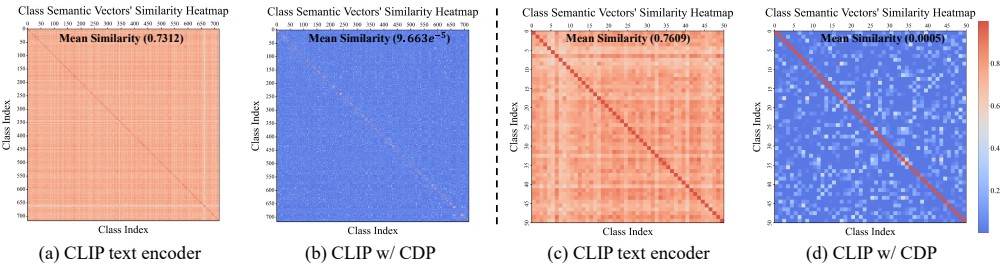

| (a) CLIP text encoder | (b) CLIP w/ CDP | (c) CLIP text encoder | (d) CLIP w/ CDP |

Figure 8: Class semantic vectors' similarity heatmaps are extracted by CLIP text encoder and CLIP with class diversity promotion on SUN (a,b) and AWA2 (c,d).

## C  GENERATIVE ZSL METHODS WITH WEAK CLASS SEMANTIC VECTORS

We provide the results of imagination-based ZSL (e.g., f-VAEGAN [54]) and induction-based generative ZSL (e.g., GenZSL) using weak class semantic vectors (e.g., CLIP text embeddings of

class names) on SUN and AWA2. Results are shown in Table 5. We find that i) the performances of f-VAEGAN drop heavily on SUN ($acc$ : $64.7\% \rightarrow 45.2\%$; $H$ : $41.3\% \rightarrow 33.3\%$) and AWA2 ($acc$ : $71.1\% \rightarrow 67.1\%$; $H$ : $63.5\% \rightarrow 59.8\%$) when it uses the weak class semantic vector rather than the strong one (e.g., expert-annotated attributes); ii) our GenZSL achieves significant performance gains over f-VAEGAN. These demonstrate that induction-based generative model is more feasible for ZSL than the imagination-based ones.

Table 5: Results of various generative ZSL methods with weak class semantic vectors on SUN and AWA2.

| Methods | SUN | | | | AWA2 | | | |
|---|---|---|---|---|---|---|---|---|
| | CZSL | GZSL | | | CZSL | GZSL | | |
| | acc | U | S | H | acc | U | S | H |
| f-VAEGAN (strong) | 64.7 | 45.1 | 38.0 | 41.3 | 71.1 | 57.6 | 70.6 | 63.5 |
| f-VAEGAN (weak) | 45.2 | 32.4 | 34.3 | 33.3 | 67.0 | 43.3 | 83.2 | 59.8 |
| GenZSL (weak) | 73.5 | 50.6 | 43.8 | 47.0 | 92.2 | 86.1 | 88.7 | 87.4 |

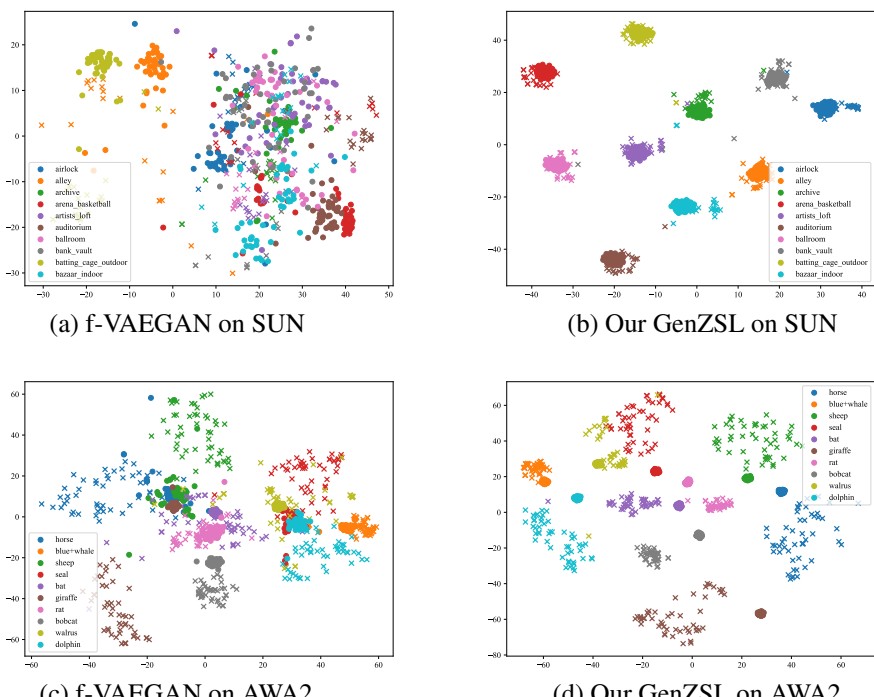

(a) f-VAEGAN on SUN        (b) Our GenZSL on SUN

(c) f-VAEGAN on AWA2        (d) Our GenZSL on AWA2

Figure 9: Qualitative evaluation with t-SNE visualization. The sample features from f-VAEGAN [54] are shown on the left, and from our GenZSL are shown on the right. We use 10 colors to denote randomly selected 10 classes from SUN (a,b) and AWA2 (c,d). The "×" and "○" are denoted as the real and synthesized sample features, respectively. The synthesized sample features and the real features distribute differently on the left while distributing similarly on the right.

## D  T-SNE VISUALIZATION ON SUN AND AWA2

As shown in Fig. 9, t-SNE visualizations of visual features learned by the f-VAEGAN [54] and our GenZSL on SUN (a,b) and AWA2 (c,d). Analogously, the visual features generated by f-VAEGAN are also far away from their corresponding real ones, and the discrimination of these real/synthesized visual features is undesirable. In contrast, our GenZSL synthesize visual features close to their corresponding real ones. As such, our GenZSL significantly improves the performances

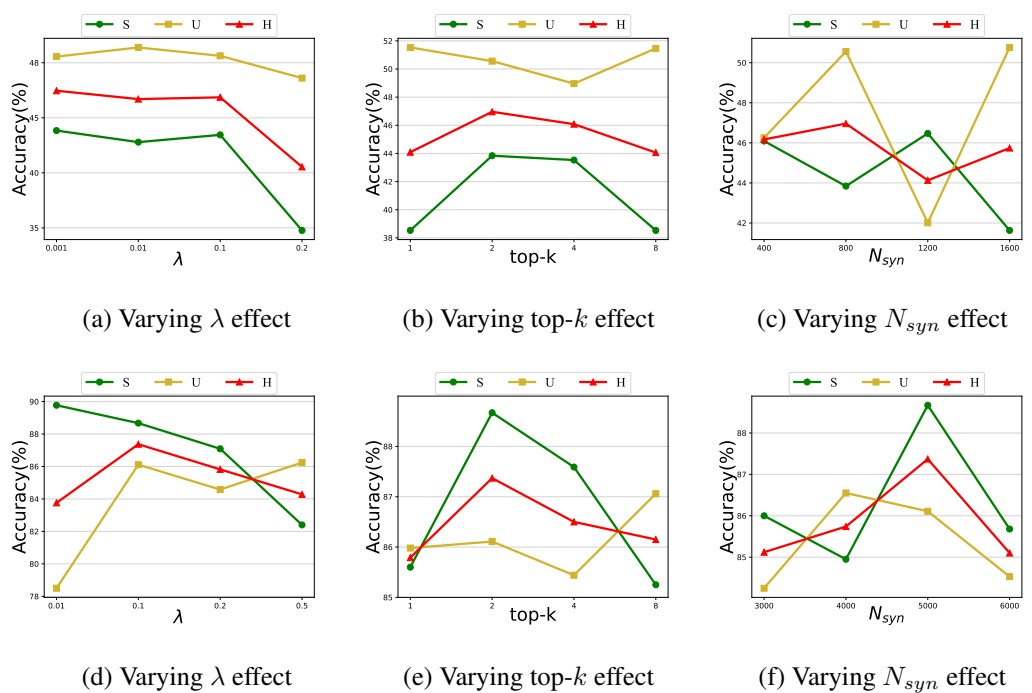

(a) Varying $\lambda$ effect  (b) Varying top-$k$ effect  (c) Varying $N_{syn}$ effect

(d) Varying $\lambda$ effect  (e) Varying top-$k$ effect  (f) Varying $N_{syn}$ effect

Figure 10: Hyper-parameter analysis. We show the performance variations loss weight $\lambda$, the number of the top referent classes top-$k$, and the number of synthesized samples of each unseen class $N_{syn}$ on SUN (a,b,c) and AWA2 (d,e,f).

of f-VAEGAN on CUB and SUN. This demonstrates that GenZSL is a effective generative ZSL model.

## E  HYPER-PARAMETER ANALYSIS ON SUN AND AWA2

We analyze the effects of different hyper-parameters of our GenZSL on SUN and AWA2 datasets. These hyper-parameters include the loss weight $\lambda$ in Eq. 5, the number of the top referent classes top-$k$, and the number of synthesized samples for each unseen class $N_{syn}$. Fig. 6 shows the GZSL performances of using different hyper-parameters. We observe that our GenZSL is robust and easy to train. We empirically set these hyper-parameters $\{\lambda, k, N_{syn}\}$ as $\{0.001, 2, 800\}$ and $\{0.1, 2, 5000\}$ for SUN and AWA2, respectively.

