# OpenReview forum: "Induction Rather Than Imagination: Generative Zero-Shot Learning Via Inductive Variational Autoencoder"
_ICLR.cc/2025/Conference — ICLR 2025 Conference Withdrawn Submission_

### Official Review · Reviewer_8b3t · 2024-10-29

**Soundness:** 2
**Presentation:** 2
**Contribution:** 2
**Rating:** 3
**Confidence:** 5

**Summary:**

This paper proposes an inductive variational auto-encoder for generative zero-shot learning, which induces new class samples from similar seen classes using weak class semantic vector derived from target class names, such as CLIP text embedding. Two strategies are incorporated to train an effective ZSL classifier. The first is employing class diversity promotion to enhance the diversity of class semantic vectors. The second is utilizing target class-guided information boosting criteria to optimize the model.

**Strengths:**

The idea of inducting new class samples for ZSL is interesting.

**Weaknesses:**

- The key method is not very clearly presented. For example, it is unclear how x^{refer} is obtained from the top k similar classes.
- The class diversity promotion is achieved by applying SVD on class semantic vectors and removing the major component e1. Will this operation potentially destroy the class semantics? Moreover, the class diversity promotion module will limit the zero-shot learning capacity of the CLIP model, since the orthogonal basis is obtained on a fixed set of classes, such as 200 classes in CUB.
- The paper fails to compare with more recent state-of-the-art methods, such as VADS [1], ZSLViT [2], and DFZSL [3], etc. Moreover, from table 4, it can be seen that the proposed method only performs comparably to CoOp + SHIP [34] baseline. For the unseen classes, the performance of GenZSL is even worse than CLIP[37] baseline, indicating that the zero-shot generalization capacity of original CLIP model is destroyed by the proposed GenZSL method.

[1] Hou, Wenjin, et al. "Visual-Augmented Dynamic Semantic Prototype for Generative Zero-Shot Learning." *Proceedings of the IEEE/CVF Conference on Computer Vision and Pattern Recognition*. 2024.

[2] Chen, Shiming, et al. "Progressive Semantic-Guided Vision Transformer for Zero-Shot Learning." *Proceedings of the IEEE/CVF Conference on Computer Vision and Pattern Recognition*. 2024.

[3] Tang, Bowen, et al. "Data-Free Generalized Zero-Shot Learning." *Proceedings of the AAAI Conference on Artificial Intelligence*. Vol. 38. No. 6. 2024.

**Questions:**

- How is x^{refer} obtained from the top k similar classes?
- Will the proposed class diversity promotion potentially destroy the class semantics?
- How to explain that the zero-shot generalization capacity of the proposed method is worse than the original CLIP model?

---

### Official Review · Reviewer_GfY1 · 2024-10-30

**Soundness:** 2
**Presentation:** 3
**Contribution:** 2
**Rating:** 3
**Confidence:** 4

**Summary:**

This paper proposes a novel generative zero-shot learning method. Unlike most existing generative ZSL methods that generates samples from Gaussian noise and class semantics, the proposed method generates unseen class samples from similar referents instead. The motivation of the proposed method is quite novel, and the performances of the proposed method surpass some existing ZSL methods on three well-studied ZSL benchmarks.

**Strengths:**

(1)	The motivation of the proposed method is quite novel and inspirational. That is, generating unseen class samples from similar references is an underexplored area in ZSL.
(2)	The proposed method is clearly presented, which is easy to follow for other researchers.
(3)	Technically, the proposed method is sound and solid.

**Weaknesses:**

(1)	The experiments are not very convincing and some experiment results are not clearly illustrated. First of all, since the authors adopt CLIP as the feature extractor, it is not clear whether the performance gain comes from the better feature or the designation of the framework (note that most existing ZSL methods use ResNet-101 features or ViT features). Second, the authors should compare with some more recently proposed and more competitive ZSL methods, e.g. VADS [1], DFCAFlow [2], ZSLViT [3], etc.. Third, Table 4 seems insufficient to validate the effectiveness of the proposed method. The proposed method adopts CLIP features and achieves better performance on seen classes and worse performances on unseen classes than the original CLIP. Such results suggest that the model might be overfitting the seen classes, which is not a desired characteristic of ZSL methods, and thus this experiment is unable to validate the superiority of the proposed method. Fourth, some experiment results are not consistent with the authors’ claim. In Section 3.1, the authors claim that different classes have high similarity under the original CLIP features. However, as shown in Figure 3 and Figure 8, the coarse-grained AwA dataset has much higher mean similarity than the fine-grained CUB and SUN datasets, which is in contrary to my instinct (similar classes, i.e. fine-grained classes should have higher similarities and coarse-grained classes should have lower similarities), and the authors did not explain such a phenomenon.
(2)	The encoder and decoder structure should be more clearly illustrated. Specifically, how to combine the class semantic with the referent feature, summation or dot production or concatenation?
(3)	Some small problems. First, the “Problem Setting” part in Section 3. The key characteristic of ZSL is that seen classes and unseen classes have no intersection. The authors did not mention this in the problem setting. Second, in lines 280-281, the definition of CZSL is wrong. I think this might just be a typo. Third, the definition of “<” and “>” are missing in Equation 4

[1] Wenjin Hou, Shiming Chen, Shuhuang Chen, Ziming Hong, Yan Wang, Xuetao Feng, Salman Khan, Fahad Shahbaz Khan, and Xinge You. Visual-augmented dynamic semantic prototype for generative zero-shot learning. In Proceedings of the IEEE/CVF Conference on Computer Vision and Pattern Recognition (CVPR), pp. 23627–23637, June 2024.
[2] Hongzu Su, Jingjing Li, Ke Lu, Lei Zhu, and Heng Tao Shen. Dual-aligned feature confusion alleviation for generalized zero-shot learning. IEEE Transactions on Circuits and Systems for Video Technology, 2023.
[3] Shiming Chen,Wenjin Hou, Salman Khan, and Fahad Shahbaz Khan. Progressive semantic-guided vision transformer for zero-shot learning. In Proceedings of the IEEE/CVF Conference on Computer Vision and Pattern Recognition, pp. 23964–23974, 2024a.

**Questions:**

(1)	The authors use CLIP as the feature extractor. However, since CLIP is trained on a very large quantity of general-purpose training samples, it is highly likely that some training samples of CLIP overlap with the unseen classes, making the comparison with existing ZSL methods unfair. To address this issue, it is necessary to conduct experiments by replacing the CLIP feature with ResNet-101 features or ViT features as in most existing ZSL methods.
(2)	The authors should compare with some more recently proposed and more competitive ZSL methods as mentioned in the “Weakness” part. If they are not directly comparable with the proposed method, the authors should explain why.
(3)	As shown in Figure 3 and Figure 8, the coarse-grained AwA dataset has higher mean similarity than the fine-grained CUB and SUN datasets, which is in contrary with my instinct (similar classes, i.e. fine-grained classes should have higher similarities and coarse-grained classes should have lower similarities), some explanations are needed.
(4)	In equation 3, how are p and q defined? Some more details are necessary, at least in the appendix.
(5)	The hyper-parameter analysis in Section 4.5 lacks insights. For example, why does too many synthesized data hurt the performance on both seen and unseen classes (Figure 6(c))? Does this phenomenon suggest that the generator is not well trained? How is X^{refer} defined when k > 2? Is it possible to adaptively select the value of k based on the similarity scores instead of setting k as a fixed hyper-parameter as in the current framework? I suggest the authors dig deeper into the results and try to explain these phenomenon to give the readers more insights into the proposed method as well as generative ZSL methods.

---

### Official Review · Reviewer_rzJg · 2024-10-31

**Soundness:** 3
**Presentation:** 4
**Contribution:** 3
**Rating:** 5
**Confidence:** 5

**Summary:**

This paper targets on zero-shot classification and learns the classifier based on the synthesized samples. The key insight is to synthesize samples of new (unseen) classes from the relevant samples of the seen classes. As such, this paper proposes to learn the inductive encoder and encoder decoder and use the pre-trained embedding spaces of CLIP to select relevant samples of each new class. However, the method design is not realistic and have assumption. In other words, I do like this concept, but the detailed methodology should be carefully curated, such as the regularization design and the selection of pre-trained models in your approach.

**Strengths:**

1.	the motivation and insight of this paper is great.
2.	Both visualization and quantitively analysis are conducted.
3.	Methodology explanation is clear

**Weaknesses:**

1.	I truly appreciate the insight about induction instead of imagination from scratch. However, the experimental setup in your evaluation is not realistic for me. In detail, you are using a  pre-trained CLIP encoder for feature extraction but does not compare with CLIP baseline. Also, CLIP has already demonstrate strong ZSL performance. Though you may claim that the vision-language models (VLM) is trained via contrastive learning to align the image and languages to facilitate the hallucination, you should still use the VLM which are not intuitively utilized for classification. After all, many distributional properties of CLIP embedding space may introduce noise in your experimental validation. For your approach, a more realistic setup will be to use encoders which are separately trained in different modalities.
2.	An important and particular question for such induction-based approach, is the balance issue. As also discussed in Fig. 6, there is always a tradeoff between classification accuracy between seen and unseen classes, and the difference is significant, partially due to overfitting. How can you mitigate such issue?
3.	Another challenge is that the theoretical validation of your indication approach. No matter it is imagination and induction, you still learn a normal distribution in the VAE. In your approach, the decoder still directly generate samples from the random noise, and I am worry about the capability or generalization of such decoder in inductive generation.
4.	(minor) The claim for the first inductive approach is too strong. There has been some research effort that I have reviewed before.
5.	(minor) Typo: What is the difference between CZSL and GZSL. I guess there is some typo in line 281. But the definition in line 153 is correct?

**Questions:**

Please address my quesrions in the weakness, in particualr the first three.

---

### Official Review · Reviewer_7rfr · 2024-11-04

**Soundness:** 2
**Presentation:** 1
**Contribution:** 2
**Rating:** 3
**Confidence:** 5

**Summary:**

This paper proposes a sample generation-based method for zero-shot classification. Instead of generating unseen category samples based on the model's "imagination," the proposed method uses relative seen categories as a reference to generate samples by "induction." A pre-trained CLIP model (both image and text encoder) is used to extract image and text features (replace the class attributes as in conventional ZSL).

**Strengths:**

The imagination of the model can cause trouble when generating unseen class samples due to uncertainty. It is good to see this work try to mitigate this issue by using samples from similar seen classes as the reference to induct samples.

**Weaknesses:**

1. f-VAEGAN is a fairly old work (CVPR2019), and ZSL (even the methods not based on CLIP) is a fast-developing area, like the works cited in Table 2. Since the proposed method is not a direct variation of f-VAEGAN nor an add-on module, f-VAEGAN should not be used as the main baseline for discussion, especially not suitable for highlighting the performance boost (like "GenZSL with significant efficacy and efficiency over f-VAEGAN ...." in the abstract). Similar to Figures 4 and 5.

2. This paper uses CLIP as a feature extractor for both vision and text input. However, most, if not all, of the listed works are based on ImageNet pre-trained ResNet or ViT. Previous works (e.g., f-vaegan and tf-vaegan) have shown that even finetuning the pre-trained model on the seen class can largely boost the ZSL performance, no need to mention that CLIP is a foundation model that is trained on 400M image-text pairs and "zero-shot" is one of the flagship features of CLIP. Previous works, like [1], also demonstrate CLIP's outstanding performance on many ZSL benchmarks, including the three datasets used in this work. Thus, the performance comparisons provided in this paper are not fair, meaningful, or provide sufficient information to evaluate the proposed work. Also, with CLIP's feature, the model greatly boosts the AWA2 dataset but only has marginal improvements on other datasets. I am curious about the reason behind this. If this major concern is completely cleared or if there is anything that I misunderstood, I would like to re-evaluate this paper.

3. The paper only contains three benchmark datasets and lacks of some commonly used benchmarks for evaluating ZSL performance, like aPY and FLO. Also, [1] includes evaluations on an extended amount of datasets that can be used as a candidate pool for dataset selection.

[1]: Wang et al., "Improving Zero-Shot Generalization for CLIP with Synthesized Prompts," ICCV2023

**Questions:**

Please refer to the weaknesses.

---

### Note · Authors · 2024-11-12

I have read and agree with the venue's withdrawal policy on behalf of myself and my co-authors.